

# APOE genotype and age modifies the correlation between cognitive status and metabolites from hippocampus by a 2D $^1$H-MRS in non-demented elders

Zhenyu Yin[1,*], Wenbo Wu[2,*], Renyuan Liu[3], Xue Liang[4],
Tingting Yu[1], Xiaoling Chen[1], Jie Feng[1], Aibin Guo[1], Yu Xie[1],
Haiyan Yang[1], Mingmin Huang[1], Chuanshuai Tian[4], Bing Zhang[4] and
Yun Xu[3]

[1] Department of Geriatrics, Affiliated Drum Tower Hospital of Nanjing University Medical School, Nanjing, Jiangsu, China
[2] Department of Neurology, Affiliated Drum Tower Hospital of Nanjing Medical University, Nanjing, Jiangsu, China
[3] Department of Neurology, Affiliated Drum Tower Hospital of Nanjing University Medical School, Nanjing, Jiangsu, China
[4] Department of Radiology, Affiliated Drum Tower Hospital of Nanjing University Medical School, Nanjing, Jiangsu, China
* These authors contributed equally to this work.

Corresponding authors
Bing Zhang,
zhangbing_nanjing@vip.163.com
Yun Xu,
xuyun20042001@aliyun.com

## ABSTRACT

**Purpose.** To examine the associations among age, Apolipoprotein E (APOE) genotype, metabolic changes in the hippocampus detected by 2D $^1$H magnetic resonance spectroscopy (MRS), and neuropsychological measures of cognition in non-demented elders.

**Materials and Methods.** We studied a cohort of 16 cognitively normal controls (CN) and 11 amnestic mild cognitive impairment (aMCI) patients between 66 and 88 years old who were genotyped for APOE genetic polymorphism. Measurements of 2D$^1$H-MRS metabolites were obtained in the hippocampus region. Adjusting by age among all subjects, the association between metabolic changes and cognitive function was measured by Spearman partial rank-order correlation. The effect of APOE status was measured by separating the subjects into APOE genotype subgroups, including the APOE$\varepsilon$4 carriers and APOE$\varepsilon$4 non-carriers.

**Results.** In contrast to the CN group matched with age, gender, and education, aMCI patients showed increased myo-inositol (mI)/Creatine (Cr) ratio only in the right hippocampus. No differences were noted on N-acetylaspartate (NAA)/Cr and mI/NAA from bilateral hippocampus, and so was mI/Cr ratio in left hippocampus between aMCI and CN. The mI/Cr ratio from the right hippocampus in non-demented elders was negatively correlated with Montreal Cognitive Assessment (MoCA) scores. Whether $\varepsilon$4 genotype or age was added as a covariate, none of the correlation effects remained significant. Additionally, adjusting for age and APOE genotype together, there was no significant correlation between them.

**Conclusion.** Since the higher mI/Cr from the right hippocampus of the patients with aMCI than those from CN, the mI/Cr could be a more specific predictor of general cognitive function in aMCI patients. There is an association between higher mI/Cr in right hippocampus and worse cognitive function for the non-demented older adults,

and the correlation could be modified by APOE status and age. That provided a window on objectively understanding the mechanism between the brain metabolites and the influence factors in non-demented elders.

## INTRODUCTION

The proton magnetic resonance spectroscopy ($^1$H MRS) is unique among diagnostic imaging modalities because the signals from several different metabolites are measured. It is a potential metabolic marker in Alzheimer's disease (AD) for both early diagnoses and evaluating treatment effects (*Graff-Radford & Kantarci, 2013*; *Murray et al., 2014*; *Tumati, Martens & Aleman, 2013*). MRS allows regional measurement of metabolites including myo-inositol (mI), choline (Cho), N-acetyl aspartate (NAA), and creatine (Cr). Cr is typically used as an internal reference to control for variability in measurement because it remains unchanged in AD (*Kantarci, 2007*). The mI/Cr ratio is associated with glia and elevated levels with glial proliferation. Glial and microglial activity raises the possibility that elevated mI represents inflammation which is an early event in the evolution of AD pathology (*Graff-Radford & Kantarci, 2013*).

However, more and more evidences showed that despite of mI initially changed in early AD, mI/Cr was not related to the pathological severity of AD and could not predict the AD progression. For example, an autopsy study showed the subjects with a low likelihood of AD and sparse neuritic plaques had higher mI/Cr ratios than the subjects with an intermediate likelihood of AD and moderate neuritic plaques (*Kantarci et al., 2008*). A longitudinal study also found the annual percent change in mI/Cr ratios was not different between amnestic subtype of MCI (aMCI)-stable and aMCI-converter who progressed to AD during follow up (*Kantarci et al., 2007*). Furthermore, the mI levels were found to be associated with general cognition (such as MMSE, MoCA) in one study (*Rose et al., 1999*), but not associated with Mini-Mental State Examination (MMSE) in another study (*Huang et al., 2001*). Therefore, the association between metabolites and cognitive measures remains controversial.

There are some factors that potentially influence the cognitive impairment. Apo lipoprotein E (APOE) genotype is the best established susceptibility gene and has been shown to influence age of onset (*Mastaglia et al., 2013*) and the underlying histopathology of AD (*Michaelson, 2014*). Carriers of the APOE$\varepsilon$4 allele have an increased risk of cognitive decline (*Kozauer et al., 2008*). Age is another main factor affecting cognitive function. A study using general linear model analysis demonstrated that older APOE$\varepsilon$4 carriers had significantly higher mI than APOE$\varepsilon$3 homozygotes in a healthy aging normal population (*Gomar et al., 2014*). But it has been previously suggested that the APOE effect dissipated for old individuals who were more than 80 year old (*Negash et al., 2009*).

To the best of our knowledge, there is still no evidence for the effect of APOE genotype and age on the relationship between metabolites and cognitive status in aMCI patients. Therefore, our primary objective was to determine the metabolic changes in the hippocampus measured by 2D $^1$H-MRS in aMCI patients compared with normal controls. We hypothesized that aMCI patients have increased mI/Cr ratio in the hippocampus compared with normal elderly. In addition, APOE genotype and age may impact the correlation between metabolites and cognitive status in aMCI patients. In this way, we might be able to elucidate the influence factors on the association between metabolites and cognitive measures.

## METHODS AND MATERIALS

### Participants

Participants were recruited at the Memory clinic of Neurology Department in Nanjing Drum Tower Hospital during a 19-month span from September 1, 2011 to April 31, 2013. The study was approved by the Medical Research Ethics Committee of Nanjing Drum Tower Hospital, Nanjing, China, and all the subjects' written informed consents were obtained before the study. Participants were classified based on the clinical criteria and results of the neropsychological tests into the aMCI and CN groups. Participants in aMCI group met the Petersen's criteria (*Petersen, 2000*; *Petersen, 2004*), which included (a) memory complaint, preferably confirmed by an informant; (b) objective memory impairment, adjusted for age and education; (c) normal or near-normal performance on general cognitive functioning and no or minimum impairment of daily life activities; (d) the Clinical Dementia Rating (CDR) score of 0.5; and (e) not meeting the criteria for dementia according to the DSM-IV (Diagnostic and Statistical Manual of Mental Disorders, 4rd edition, revised). Patients with aMCI were diagnosed by experienced neurologists. Cognitive normal controls (CN) were identified as individuals who (a) had no cognitive complaints, (b) had a normal level of clinical rating scales, and (c) had no evidence of any abnormality examined by a conventional MRI.

Subjects were excluded if they met the following clinical characteristics: (a) those who have a clear history of stroke; (b) severe depression that led to aMCI (Hamilton Depression Rating Scale score > 24 points)/vascular dementia (Hachinski scores $\geq$7); (c) other nervous system diseases, which can cause cognitive impairment (such as brain tumors, Parkinson's disease, encephalitis, and epilepsy); (d) cognitive impairment caused by traumatic brain injury; (e) other systemic diseases, which can cause cognitive impairment, such as thyroid dysfunction, severe anemia, syphilis, and HIV; and (f) a history of psychosis or congenital mental growth retardation.

### Cognitive assessment

All of subjects were evaluated using a standardized clinical evaluation protocol, which included Montreal Cognitive Assessment (MoCA), Mini-Mental State Examination (MMSE), Clinical Dementia Rating (CDR), Activity of Daily Living Scale (ADL), Hachinski Ischemic Scaling (HIS) and Hamilton Depression Scale (HAMD) to exclude

vascular cognitive impairment and depression state. Memory and executive function was assessed through ADAS-cog test. Wechisler Adult Intelligence Scale including digital symbol and digital span were also tested for all participants.

## Genotpying

Genotyping was conducted using Polymorphic DNA at two loci in APOE exon 4 by amplicon sequencing methods to produce small polymerase chain reaction products that serve as the templates for bidirectional sequencing. Participants with APOE genotype 2/2, 2/3 were labeled$\varepsilon$2 carriers, genotype 3/3 was labeled$\varepsilon$3 homozygote, genotypes 3/4 and 4/4 were labeled $\varepsilon$4 carriers.

## Proton magnetic resonance spectroscopy

All MRI examinations were performed at 3 T using an 8-channel phased array head coil (Achieva 3.0T TX dual-source parallel RF excitation and transmission technology, Philips Medical Systems, The Netherlands).

A three-dimensional turbo fast echo (3D-TFE) $T_1$WI sequence on sagittal view with repetition time (TR)/echo time (TE) $= 9.8/4.6$ ms; flip angle $= 8°$; field of view (FOV) $= 256 \times 256$ mm$^2$, and a slice thickness of 1 mm was performed for reformatting transverse images. Then, the transverse image of bilateral hippocampi as the main reference image was obtained by reformatting the 3D-TFE $T_1$WI image from Philips Achieva 3.0T MR Extended Workspace (EWS) for localizing the $^1$H-MRS sequence on hippocampus. A two-dimensional Point-resolved spectroscopy (2D-PRESS) pulse sequence with TR/TE $= 2,000/32$ ms, flip angle $= 90°$, FOV $= 100 \times 100$ mm, acquisition voxel $= 12 \times 12$ mm, reconstruction voxel $= 4 \times 4$ mm and thickness $= 8$ mm was performed for each side of the hippocampus, respectively. The VOI (voxel of interest, $64 \times 32$ mm in this study) of 2D-PRESS was located on the transverse image of bilateral hippocampi, and its long axis is parallel with the hippocampus referenced by sagittal $3DT_1W$ image, with the caution of avoiding the surrounding bone, air and fat. We selected the hippocampus voxel (18~27) in the effective voxel based on the hippocampal anatomy actual form. A voxel was considered eligible whenever more than two-thirds of its area was located within hippocampus area. Finally, the average magnitude spectra per voxel were computed by summation of the voxel spectra from the defined volume of interest (VOI) and division by the number of voxels (*Shen et al., 2009*; *Weis et al., 2014*). The chemical shift distance was considered in the localization process by double-checking the chemical shift voxels of NAA and mI, respectively (Fig. 1). The MRS acquisition time was about 6 min and 6 s.

The MRS metabolite ratios were determined for NAA/Cr, mI/Cr and mI/NAA. Ratios were quantified automatically with Philips WorkStation software (Extended Workspace, EWS). The measured Cr peak includes both the metabolites Cr and phosphocreatine and is thought to be a reliable marker of brain energy metabolism, often used as a relatively stable reference level (*Kantarci, 2007*).

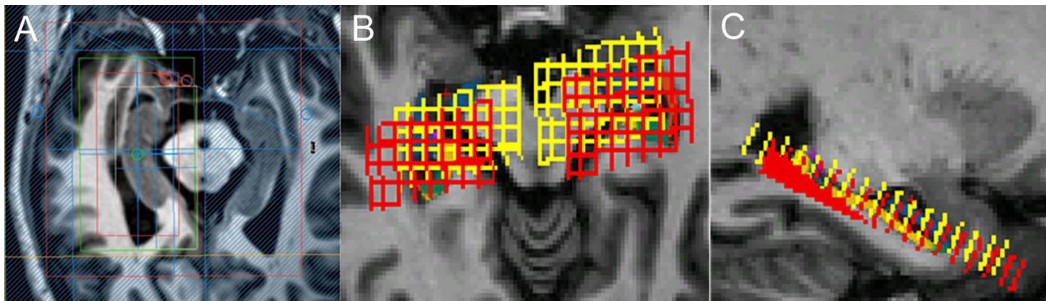

**Figure 1** **The localization of MRS on hippocampus.** (A) Different VOIs for NAA (red) and mI (white) for localization. Shimming (green) and REST slabs (blue) are also shown above. (B), (C) Different VOIs for NAA (red) and mI (yellow) co-registered in FreeSurfer native space.

## Statistical analyses

Statistical analysis was performed by software (SPSS version 16.0; SPSS, Chicago, Ilinois) for demographic and clinical data. For our primary analysis, we summarize associations between cognition and $^1$H-MRS metabolite ratios using Spearman partial rank-order correlations which we denote by "partial $r_s$." Due to the limitation of our sample size, we used partial $r_s$ to quantify associations since it is a nonparametric correlation between 2 variables which does not assume normally distributed data and therefore is preferable in this situation. Next, we report partial Spearman rank-order correlations adjusting for age, gender, and education among all subjects. The effect of APOE genotype on the association between cognitive function and $^1$H-MRS metabolite ratios was adjusted by APOE genotype subgroups. We separated the APOE$\varepsilon$4 carriers ($\varepsilon$3$\varepsilon$4 and $\varepsilon$4$\varepsilon$4) and APOE$\varepsilon$4 non-carriers ($\varepsilon$2$\varepsilon$2, $\varepsilon$2$\varepsilon$3 and APOE$\varepsilon$3 homozygotes) into two subgroups.

In summary, we tested for associations using Spearman partial rank-order correlations in which we report 4 p values: (1) the significance of associations between cognition and $^1$H-MRS metabolite ratios; (2) whether adding age to this model is significant; (3) whether adding APOE to this model is significant; and (4) whether adding age and APOE to this model is significant.

## RESULTS

### Subjects

Characteristics of subjects are described in Table 1. Subjects with Fazekas III and high Hachinski scores ($\geq$7) were excluded in this study in order to reduce and isolate the effect of vascular dementia. There was no significant difference between aMCI ($N = 11$; age: 79 $\pm$ 6.5, 69–88; 2 female and 9 male; education years: 14 $\pm$ 2.9, 9–16) and CN ($N = 16$; age: 74 $\pm$ 5.5, 66–83; 4 female and 12 male; education years: 15 $\pm$ 1.0, 12–16) in age ($p = 0.056$), gender ($p = 0.675$), and education ($p = 0.342$) by Mann–Whitney U test. Significantly lower cognitive performances in the aMCI (MMSE: 25.7 $\pm$ 3.5, 20–30; MoCA: 21.0 $\pm$ 3.6, 14–26; Hachinski: 3.64 $\pm$ 2.42, 0–6; ADAS-cog: 10.2 $\pm$ 6.27, 2–23) compared with the CN (MMSE: 28.8 $\pm$ 1.7, 24–30; MoCA: 26.9 $\pm$ 1.5, 24–30; Hachinski: 1.81 $\pm$ 1.27, 0–4; ADAS-cog: 5.5 $\pm$ 2.98, 1–14) were observed (all p values <0.05).

**Table 1** Demographic data and patient characteristics.

| | CN $n = 16$ | aMCI $n = 11$ | $p$ | $Z/\chi^2$ |
|---|---|---|---|---|
| No. female (%) | 4 (25) | 2 (18.2) | 0.675 | 0.175 |
| Age, $y$ | 74 (66–83) | 79 (69–88) | 0.056 | −1.929 |
| Education, $y$ | 15 (12–16) | 14 (9–16) | 0.342 | 0.119 |
| No. APOE$\varepsilon$2 carriers (%) | 1 (6.3) | 2 (18.2) | | |
| No. APOE$\varepsilon$3 homozygotes (%) | 12 (75) | 7 (63.6) | 0.620 | 0.956 |
| No.APOE$\varepsilon$4 carriers (%) | 3 (18.8) | 2 (18.2) | | |
| No. FazekasII(%) | 3 (18.8) | 3 (27.3) | 0.601 | 0.274 |
| MMSE | 28.8 (24–30) | 25.7 (20–30) | **0.013**[*] | −2.520 |
| MoCA | 26.9 (24–30) | 21.0 (14–26) | **0.000**[*] | −4.039 |
| ADL | 20.3 (20–23) | 22.8 (20–42) | 0.162 | −1.862 |
| Hachinski | 1.8 (0–4) | 3.6 (0–6) | **0.026**[*] | −2.257 |
| HAMD | 2.5 (0–19) | 1.9 (0–7) | 0.645 | −0.525 |
| CDR | 0.0 (0.0–0.0) | 0.5 (0.5–0.5) | **0.001**[*] | 11.221 |
| ADAS-cog | 5.5 (1–14) | 10.2 (2–23) | **0.023**[*] | −2.278 |
| Wechsler | 32.0 (20–52) | 25.0 (8–57) | 0.099 | −1.680 |
| Digit span forward | 7.8 (6–9) | 6.9 (4–10) | 0.134 | −1.540 |
| Digit span backward | 5.1 (3–8) | 4.2 (2–7) | 0.162 | −1.442 |

**Notes.**

CN, cognitively normal control; aMCI, amnestic Mild Cognitive Impairment; MMSE, Mini Mental State Examination; MoCA, Montreal cognitive assessment; HAMD, Hamilton Depression Scale; CDR, Clinical Dementia Rating; ADL, Activities of daily living; ADAS-Cog, Alzheimer's Disease Assessment Scale-Cognitive subscale.

[*] aMCI is significantly different from CN at $p < 0.05$.

Median (interquartile range) was reported for continuous variables.

## $^1$H-MRS metabolite ratios

The characteristics of $^1$H-MRS metabolite ratios of bilateral hippocampi, including NAA/Cr, mI/Cr and mI/NAA, are described in Table 2. We treated $^1$H-MRS metabolite ratio as a continuous variable in all analysis. Increased mI/Cr in right hippocampus of aMCI subjects compared with CN was found by Mann–Whitney $U$ test ($p = 0.050$) (Fig. 2). However, no significant difference in NAA/Cr in bilateral hippocampi and mI/Cr in left hippocampus was found (all $p$ values >0.05).

## Correlation between $^1$H-MRS metabolite ratios and cognitive function

The correlations between $^1$H-MRS metabolite ratios, age and cognition scores were measured by Spearman partial rank-order correlations as described in Table 3. Only mI/Cr in right hippocampus was enrolled in the association analysis since it was the only metabolite ratio appeared significant difference between groups. The same principal was applied to cognitive rating scales, as only five of them are shown in Table 3. Age is a major factor of cognition decline and AD neuropathology and therefore was also included in this analysis.

Higher mI/Cr in right hippocampus was associated with worse overall cognitive performance, including MMSE, MoCA, Hachinski, CDR and ADAS-cog. However, only

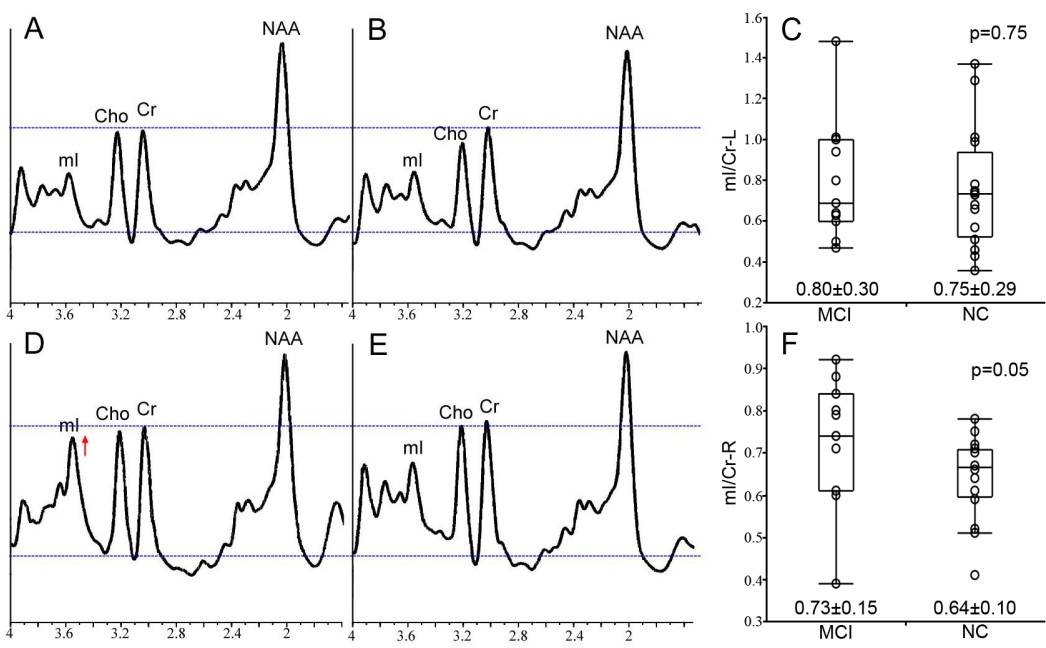

**Figure 2 The MRS result from bilateral hippocampus.** The representative example MRS resulting from the left hippocampus of the aMCI and CN subject is respectively shown in (A) and (B), and the group statistical characteristics are shown in (C). The MRS resulting from right hippocampus are showed in (D–F). All spectra are scaled to the height of the reference peak Cr, shown with a dotted line. Spectra from the right hippocampus demonstrates the increased mI/Cr in a patient with aMCI (D) compared with CN (E) at $p = 0.05$ (F).

**Table 2 MRS metabolite ratios in median (interquartile range) in aMCI group compared with CN.**

|            | CN $n = 16$        | aMCI $n = 11$      | $p$       | $Z$     |
|------------|--------------------|--------------------|-----------|---------|
| NAA/Cr R   | 2.38 (1.68–3.92)   | 2.60 (2.00–3.43)   | 0.134     | −1.530  |
| NAA/Cr L   | 2.06 (0.13–3.26)   | 2.22 (1.40–3.37)   | 0.680     | −0.444  |
| mI/Cr R    | 0.64 (0.41–0.78)   | 0.72 (0.39–0.92)   | **0.050**[*] | −1.977  |
| mI/Cr L    | 0.75 (0.36–1.37)   | 0.79 (0.47–1.48)   | 0.753     | −0.321  |
| mI/NAA R   | 0.28 (0.16–0.40)   | 0.28 (0.12–0.46)   | 0.942     | −0.099  |
| mI/NAA L   | 0.95 (0.13–10.31)  | 0.35 (0.24–0.57)   | 0.753     | −0.321  |

**Notes.**
Median (interquartile range) was reported for continuous variables.
$P$ values for group comparison are from Mann-Whitney Tests.
R, right hippocampus; L, left hippocampus; NAA, N-acetyl aspartate; mI, myo-inositol; Cr, creatine.
[*] is significantly different between groups at $p \leq 0.05$.

the correlation between mI/Cr and MoCA is significant ($r_s = -0.403$, $p = 0.025$). Age was significantly associated with MMSE ($r_s = -0.508$, $p = 0.007$), MoCA ($r_s = -0.407$, $p = 0.035$) and ADAS-cog ($r_s = 0.469$, $p = 0.014$). However, no correlation between age and mI/Cr in right hippocampus was found ($p = 0.678$).

**Table 3 The statistically significant correlations between clinical rating scales, age and the mI/Cr ratios from the right hippocampus.**

|  | MMSE | | MoCA | | Hachinski | | ADAS-cog | | CDR | | Age | |
| --- | --- | --- | --- | --- | --- | --- | --- | --- | --- | --- | --- | --- |
|  | Sig | $r_s$ | Sig | $r_s$ | Sig | $r_s$ | Sig | $r_s$ | Sig | $r_s$ | Sig | $r_s$ |
| mI/Cr R | 0.953 | −0.012 | **0.025**[*] | −0.431 | 0.719 | −0.073 | 0.522 | 0.129 | 0.165 | 0.275 | 0.678 | −0.084 |
| Age | **0.007**[*] | −0.508 | **0.035**[*] | −0.407 | 0.081 | 0.342 | **0.014**[*] | 0.469 | 0.109 | 0.315 | – | – |

Notes.
Only statistically different clinical rating scales between groups were enrolled in this correlations analysis.
MMSE, Mini Mental State Examination; MoCA, Montreal cognitive assessment; ADAS-Cog, Alzheimer's Disease Assessment Scale-Cognitive subscale; CDR, Clinical Dementia Rating; R, right hippocampus; mI, myo-inositol; Cr, creatine.
[*] is significantly different between groups at $p \leq 0.05$.

**Table 4 The effects of APOE and age on the association between mI/Cr ratios and clinical rating scale (the scores of MoCA).**

|  |  | Sig | $r_s$ |
| --- | --- | --- | --- |
| mI/Cr R ∗ MoCA |  | **0.025**[*] | −0.431 |
| (controlling for age) |  | 0.073 | −0.358 |
| (group by APOE genotype) | non-ε4 carrier | 0.152 | −0.316 |
|  | ε4 carrier | 0.219 | −0.667 |
| (group by APOE genotype and controlling for age) | non-ε4 carrier | 0.115 | −0.354 |
|  | ε4 carrier | 0.138 | −0.862 |

Notes.
MoCA, Montreal cognitive assessment; R, right hippocampus; mI, myo-inositol; Cr, creatine.
[*] is significantly different between groups at $p \leq 0.05$.

## Effect of APOEε4 allele and age on the association between cognitive function and [1]H-MRS metabolite ratios

Since there was correlation between MoCA and mI/Cr, we further analyze the effect of APOE and age on this association by partial correlation. No correlation was found between cognition and mI/Cr in right hippocampus after controlling for age ($r_s = -0.358$, $p = 0.073$) as described in Table 4, indicating that there was an interaction with age for the association between cognitive function and [1]H-MRS metabolite ratios.

The effect of APOE status was measured by separating the subjects into APOE genotype subgroups, the APOEε4 carriers ($n = 5$, allele ε3ε4 and ε4ε4) and APOEε4 non-carriers ($n = 22$, allele ε2ε3 and ε3ε3). We did not find any differences in mI/Cr (Fig. 3A) and cognition scores (Fig. 3B) between the 2 groups. No correlation was found between cognition and mI/Cr in right hippocampus in both APOEε4 carriers ($r_s = -0.667$, $p = 0.219$) and APOEε4 non-carriers ($r_s = -0.316$, $p = 0.152$) (Fig. 4). Furthermore, there was no correlation between cognitive function and [1]H-MRS metabolite ratios in both APOEε4 carriers ($r_s = -0.862$, $p = 0.138$) and APOEε4 non-carriers ($r_s = -0.354$, $p = 0.115$) after controlling for age, indicating that the APOE status and age might influence the correlation between cognition and [1]H-MRS metabolite ratios as shown in Table 4.

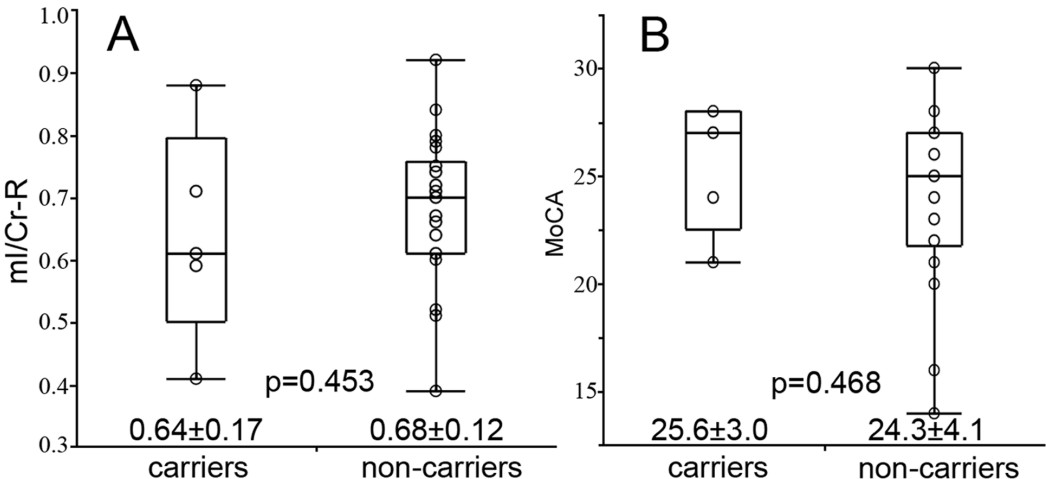

**Figure 3** **The difference of mI/Cr and MoCA between APOE$\varepsilon$4 carriers and non-carriers.** The statistical characteristics of mI/Cr from right hippocampus of APOE $\varepsilon$4 carriers and non-carriers are shown in (A). The cognitive performances of APOE $\varepsilon$4 carriers and non-carriers are shown in (B).

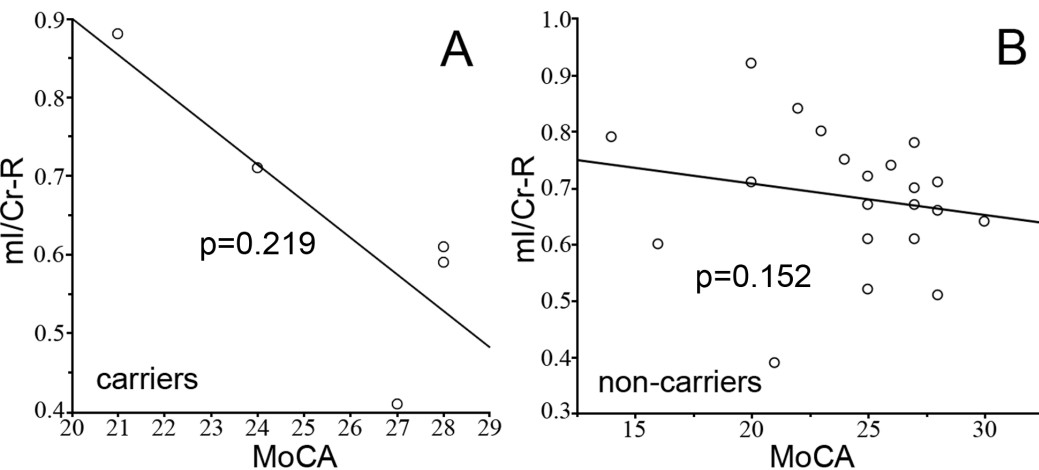

**Figure 4** **The correlation between mI/Cr and MoCA.** The correlation between mI/Cr from right hippocampus and MoCA are shown in APOE $\varepsilon$4 carriers (A) and non-carriers (B), respectively.

## DISCUSSION

Findings of this study indicated that the glial activity marker mI/Cr from the right hippocampus increased in patients with aMCI compared with those from normal elderly. Furthermore, there is an association between higher mI/Cr in right hippocampus and worse cognitive function in non-demented older adults and this relationship could be modified by APOE status and age.

The neurofibrillary tangles pathology of AD follow a typical progression from limbic to neocortical regions as AD advances (*Braak & Braak, 1991*). A similar temporal course of changes in these metabolites is seen with aMCI, a prodromal stage of AD (*Knopman, 2013*; *Petersen et al., 2009*). Therefore, it is reasonable that we selected to measure metabolites in bilateral hippocampus known as the earliest involved limbic region in aMCI patients.

Our findings are consistent with prior studies showing that aMCI patients have elevated myo-inositol levels in the right hippocampus, which is associated with glial proliferation (*Miller et al., 1993*). We did not find the changes in NAA/Cr from the hippocampus in our subject sample, which supports the hypothesis that the elevation of mI/Cr ratio precedes the decrease in NAA/Cr ratio during the progression of aMCI, mild AD and AD (*Kantarci et al., 2000*). This finding is consistent with a previous $^1$H-MRS study in aMCI (*Kantarci et al., 2000*; *Kantarci et al., 2007*; *Zhang et al., 2009*), mild AD (*Huang et al., 2001*) and in presymptomatic carriers of the familial AD mutations (*Godbolt et al., 2006*) which suggested that mI/Cr elevation is an early event in the progression of AD pathology.

In an imaging–autopsy correlation study, the antemortem mI/Cr levels correlated with the density of neuritic plaques in subjects at autopsy (*Kantarci et al., 2008*). If mI/Cr is a marker associated with the amyloid pathology of AD, then the higher mI/Cr will be expected to relate with the worse cognitive performance. In our study, mI/Cr ratios negatively correlated with the general cognition status (MoCA scores), which is in consistent with former studies (*Jessen et al., 2000*; *Kantarci et al., 2002*). Associations were present in regions where there was significant amyloid deposition, such as hippocampus. Based on this, we might expect to find an association between higher mI/Cr and worse cognitive performance in bilateral hippocampus. However, we only found the mI/Cr changed in right hippocampus, not in the left. There were published researches indicating that the rightward asymmetry of hippocampal connectivity observed in elderly controls was diminished in AD patients (*Wang et al., 2006*). Based on this potentially asymmetric hippocampus function, it is reasonable that the metabolites might change in different rate or extent between bilateral hippocampus. However, due to the small sample size limitation in our study, there is not sufficient evidence to support the hippocampus involved asymmetrically. The absence of this expected correlation requires further large sample and longitudinal investigation.

The association between the brain metabolites and cognition in old adults appear to be multifactorial, such as genotype and age. In humans, there are three common alleles of the APOE gene, numbered 2, 3 and 4. In line with prior research (*Munoz & Feldman, 2000*), APOE$\varepsilon$4 is disproportionately represented in our patient sample (16.6% of the patient sample was $\varepsilon$4+ compared to 11.5% of the control sample). Increasing researches showed that the APOE$\varepsilon$4 allele is associated with a greatly increased risk of AD (*Strittmatter et al., 1993*). However, the function of the APOE protein, and its relationship with metabolites in the brain remains mostly unknown. Although one study showed that myo-inositol were significantly increased in APOE$\varepsilon$4 carriers in a healthy aging normal population (*Gomar et al., 2014*), there is still no evidence in aMCI carriers. In our cohort, the presence of an APOE$\varepsilon$4 allele influences the relationship between mI and cognitive function in non-demented people. We dichotomized the non-demented subjects into APOE$\varepsilon$4 carriers and non-carriers groups, and did not find any differences in mI/Cr and cognition scores between the 2 groups. Within each group, there was no correlation between the mI/Cr ratio and MoCA scores, indicating APOE as the mediator modified this relationship, since there was correlation between metabolites and cognition before separating the

genotyping. Further, we note that the mI/Cr trends to decrease and the cognition scores trends to increase with the presence of an APOE$\varepsilon$4 allele. There was relatively little cognitive disturbance in non-demented APOE$\varepsilon$4 carriers in this small cohort, suggesting the influence of additional mediators such as potential compensatory progress in aMCI, which need further confirmation in the future (*Kantarci et al., 2012*).

Moreover, age is the main factors affecting cognitive function. But the age effects on $^1$H-MRS metabolites have been inconsistently described, have been small effect, or have been dependent on metabolite and voxel placement (*Haga et al., 2009*). In our sample, we found the negative correlation between age and MoCA, indicating cognition decline in old people, which is in line with other studies (*Ashworth et al., 2014*; *Yu et al., 2014*). While using controlling age as the covariates, the correlation between mI/Cr ratio and MoCA scores disappeared. Based on this result, age is another main factor affecting the correlation between metabolites and cognitive function. Further large cohort is required to confirm the age effect on the association.

Several other technique points about our study may be worth considering. One potential concern is the metabolites from hippocampus measuring by the 2D $^1$H-MRS. Compared with single voxel MRS, the surrounding tissue (i.e., fat, bone, air and cerebral spinal fluid) can be relatively easily avoided by selecting the usable voxels from the multi voxels in the field of volume, which making the metabolites from the relative pure hippocampus tissue. In addition, 2D MRS had more sampling points, and we could get more data. The mean value was more accurate and persuasive. Secondly, the metabolite measurements in this study were obtained from right and left hippocampus, respectively, which improved the quality of the shimming in the field of volume for each of the hippocampus. Thirdly, we controlled the data quality from multi-voxel chemical shift imaging $^1$H-MRS following several technical aspects: (1) using short echo time (TE = 32 ms) in $^1$H-MRS acquisitions was critical to this study, because mI has a relatively short transverse relaxation time and can be quantified only at short echo times; (2) Because the various metabolites in the tissue of interest process are at different frequencies, as a result, the selected volume will be different for the different metabolites, i.e., chemical shift displacement. Therefore, the chemical shift distance were considered in to the localization process by double-checked the chemical shift voxels of NAA and mI, respectively.

A limitation of our study is the insufficient follow-up on the cohort and the small sample size we scanned during the last 2 years. Overall, our *in vivo* $^1$H-MRS data are consistent with the biochemical changes underlying the known pathology in each of the related studies. Moreover, due to the small sample size limitation, we did not separate the cohort into aMCI subtype with different cognitive domain impairment. Amnestic MCI has higher risk at AD progression, and it is expected to have higher mI/Cr ratio and stronger correlation with APOE status.

In summary, we found mI/Cr ratios from right hippocampus increased in aMCI, suggesting glial proliferation changes could be a more specific predictor of general cognitive function in aMCI patients. Furthermore, we demonstrated that the APOE status and age modified the associations between mI/Cr from right hippocampus measured by

2D $^1$H-MRS and cognitive function in non-demented subjects. The findings of this study are of significance in further understanding the influence of changes of brain metabolites on cognitive functions, where age and APOE genotype should be taken into consideration.

## ACKNOWLEDGEMENT

The statistical analysis was supported by Biyun Xu, a statistician belonging to the Affiliated Drum Tower Hospital of Nanjing University Medical School.

### Funding

This work was supported by the National Science Foundation of China, (2014-2016, 81300925, BZ), the Provincial Natural Science Foundation of Jiangsu (2014-2016, BK20131085, BZ), and Jiangsu Province Medical Key talent people and "the 12th five years plan for China development" (2011-2-16, RC2011013, BZ). The funders had no role in study design, data collection and analysis, decision to publish, or preparation of the manuscript.

### Grant Disclosures

The following grant information was disclosed by the authors:
National Science Foundation of China: 2014-2016, 81300925.
Provincial Natural Science Foundation of Jiangsu: 2014-2016, BK20131085.
Jiangsu Province Medical Key.
12th five years plan for China development: 2011-2-16, RC2011013.

### Competing Interests

The authors declare there are no competing interests.

### Author Contributions

- Zhenyu Yin conceived and designed the experiments, performed the experiments, analyzed the data, contributed reagents/materials/analysis tools, wrote the paper, prepared figures and/or tables.
- Wenbo Wu performed the experiments, analyzed the data, contributed reagents/materials/analysis tools, wrote the paper, prepared figures and/or tables.
- Renyuan Liu performed the experiments, analyzed the data.
- Xue Liang performed the experiments, analyzed the data, prepared figures and/or tables.
- Tingting Yu, Xiaoling Chen, Jie Feng, Aibin Guo, Yu Xie, Haiyan Yang and Mingmin Huang contributed reagents/materials/analysis tools.
- Chuanshuai Tian performed the experiments.
- Bing Zhang and Yun Xu conceived and designed the experiments, reviewed drafts of the paper.

## Human Ethics

The following information was supplied relating to ethical approvals (i.e., approving body and any reference numbers):

The study has been approved by the Medical Research Ethics Committee of Nanjing Drum Tower Hospital, Nanjing, China—#NJMU20113343.

## Supplemental Information

Supplemental information for this article can be found online at http://dx.doi.org/10.7717/peerj.1202#supplemental-information.

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
