# Peer review of "APOE genotype and age modifies the correlation between cognitive status and metabolites from hippocampus by a 2D 1H-MRS in non-demented elders"

_PeerJ, doi:10.7717/peerj.1202_

## Round 0.1 · original submission · Major Revisions

Please revise the paper according to the comments of the four reviewers.

Reviewer 1 ·

Basic reporting

It is recommended to polish the manuscript by a native English speaker to make it clearer and more fluent.

Experimental design

No Comments

Validity of the findings

No Comments

Additional comments

This study revealed the relationship between brain metabolites and cognitive functions. Furthermore, the authors found the effects of age and APOE genotype on this relationship. The finding of the study is of significance in further understanding the influence of changes of brain metabolites on cognitive functions, where age and APOE genotype should be taken into consideration. In the last two sentences of the Conclusion( in Abstract), there is logically uncertain when using "non-demented older adults" before and " aMCI patients" later.

Reviewer 2 ·

Basic reporting

No Comments.

Experimental design

No comments

Validity of the findings

Concern with statistical analysis -especially if enough sample size is presented.

Additional comments

This manuscript investigated the association between the cognitive status and hippocampus metabolites for both normal controls and MCI patients.
The main concern that I have is with the statistical analysis, especially the number of sample size. The study was conducted with a cohort of 16 normal controls and 11 MCI patients. First test showed the correlation between the metabolites and the cognitive status, which is OK considering they are using Spearman’s rank correlation test. It demonstrated the mI/cr ratio could be a potential biomarker for aMCI. The following hypothesis tests are problematic. Not sure what we can get from the correlation between the cognition score and metabolites ratio. The claim in Page 11, line 212 has no statistical basis. Especially when the correlations of the scores with age are better than the mI/cr ratio, does that mean age is a better marker for cognition status? In the latter statistical analysis, considering there were only 5 patients in the carriers group, disproportionally versus 22 patients in the non-carriers group, it is hard to say if the statistical results are valid enough to support the claim that mI/Cr ratio doesn’t correlate with the APOE GENE. The claim for the trend for mI/Cr decrease with MoCA score was also weak due to the very limited 5 data points. The limited sample size was further subgrouped by age, which weakens the power of the test even more. I strongly recommend the authors discuss the statistics with a statistician and calculate the effective sample size for every test.
If possible (with larger cohort size), I would conduct the test with separated groups of normal controls and groups of aMCI patients.
Other major points include:
1. It is not clear why bilateral MI/cr levels were selected. I am curious to know how the overall level (left + right) of mI/cr level is associated with the cognitive status.
2. In the method section, there is no mentioning about the Mann-Whitney U test and Chi-square test, but only Spearman partial rank-order correlations.
3. It is not clear how the age group were controlled considering the range of age is from .

Other minor points include:
1. line 82, spell out “APOE”
2. line 103, 2-year span is only 19 months?
3. line 117, the sentence was a repeat from line 105.
4. line 143, is it a head coil?
5. line 148: it is not “reconstructing”, but “reformatting”
6. line 151 was a repeat from line 149
7. lie 165: need reference
8. Several grammatical errors
a. Abstract: line 27…. association “among” age, …
b. Abstract: line 34, Adjusted “by” age….
c. line 71: although changed to “despite of”, remove “, but”
d. line 75: remove “And”, line 76 study “also”
e. line 81: association “between” instead of “of”
f. line 86: A study “using” instead of “by”
g. line 89: rewrite the part after “that”
h. line 89: explain “old-old” individual
i. line 127: remove “by”
j. line 136: replace “by” by “using”
k. line 138: insert “the” after “as”
l. line 143: add “All” at the beginning
m. line 159: remove “to”
n. line 160: change “double-checked” to “double-checking”
o. line 161: change “is” to “was”
p. line 180: change “test” to “tested”
q. line 191: change “cognition” to “cognitive”
r. line 221: change “analysis” to “analyze”
s. line 263: add “is” before “in consistent”
t. line 307: change “relative” to “relatively” or just remove it.
u. line 299: change “when” to “using”

Reviewer 3 ·

Basic reporting

Please see my general comments and specific comments.

Experimental design

Please see my general comments and specific comments.

Validity of the findings

Please see my general comments and specific comments.

Additional comments

General Comments for the Author

RE: APOE genotype and age modifies the correlation between cognitive status and metabolites from hippocampus by a 2D 1HMRS in non-demented elders

This study was to evaluate the associations of age, Apolipoprotein E (APOE) genotype and metabolic changes in the hippocampus to the cognition measurements in 11 non-demented elders (aMCI) and 16 normal controls. The normal control group were age-, gender-, and education-matched to the aMCI patient group.

Among the three investigated metabolic measurements by MRS (NAA/Cr, ml/Cr and ml/NAA), only the ml/Cr in the right hippocampus were found significantly different between the normal control group and the aMCI group (p = 0.05), and the aMCI group had higher ml/Cr than the normal group; ml/Cr in the left hippocampus was not different between the control group and the aMCI group; NAA/Cr and ml/NAA in the bilateral hippocampi were not different between the control group and the aMCI group.

As far as the 4 cognitive assessments (MMSE, MoCA, Hachinski, and ADAS-cog), only MoCA was significantly correlated to ml/Cr (right hippocampus), p =0.025; three of them (MMSE, MoCA, and ADAS-cog) showed significant correlations to age, p < 0.035, and there were no correlation between age and ml/Cr (right hippocampus), p = 0.678.

Finally, the study did not find the effect of genotype APOE on either the metabolic measurement of ml/Cr, or the cognitive assessment of MoCA; further, when APOE and age were adjusted, the significant correlation between ml/Cr (right hippocampus) and MoCA (p = 0.025) disappeared, suggesting there were strong interactions between ml/Cr, APOE and age.

Previous in vivo studies found the asymmetric hippocampal connectivity in normal subjects disappeared in AD (Neuroimage 31:496-504); an autopsy study found higher ml/Cr in hippocampus in AD when compared with low likelihood AD (Radiology 248:210-220). The current in vivo study found, interestingly, that the ml/Cr in right hippocampus but not in left hippocampus was higher even in aMCI patients when compared with normal subjects, and there were strong interactions between ml/Cr, APOE and age in terms of their correlations to the cognitive assessment, MoCA.

I am curious about the discussion of asymmetric hippocampal connectivity (Neuroimage 31:496-504), where only right-handed subjects were recruited. It is possible that most subjects in this study were right-handed. In a future study, if more left-handed subjects (aMCI and normal) could be recruited, then could the pattern of ml/Cr in left and right hippocampi be reversed from the current findings?

Overall, this study was very well organized, and documented. However, there are some minor English grammar errors that should be corrected.

The strength of this study is that all the participants (normal controls and aMCI patients) underwent full clinical cognitive assessments, genotyping and MRS for brain metabolic measurements.

The weakness of this study is its relatively small sample size; future studies with large sample size are desired for confirming the findings.

Specific Comments:
1. Page 1 Line 1, Please specify what the APOE stands for in the title, because this is first time APOE is mentioned in the manuscript.
2. Because an important part of the study is the normal control group, please add “and matched normal controls” at the end of the title.
3. Page 2 line 43, “... negatively was correlated with …” should be “was negatively correlated with …”.
4. Page 4 line 80, please add “study” after “… in another”.
5. Page 7 line 130. What does “… to exclude vascular cognitive impairment and depression state”? Was this what you had done in the study? If yes, please add this to the excluding criteria in the participant section (page 6 line 118).
6. Page 9 line 161, “… is about 6min6s” should be “was 6 minutes and 6 seconds”.
7. Page 9 line 179, ‘… into different subgroups’, did this mean “… into two subgroups”? Page 9 line line 181, there should be a space between “4” and “p”.
8. Page 10 line 186, “subjects with Fazekas III and high Hachinski scores” were excluded in this study. Again this statement was not mentioned in the excluding criteria in the method (participant) section (page 6 line 118), please add.
9. Page 13 line 257, “…that suggested that ml/Cr …” should be “.., which suggested that ml/Cr …”.
10. Page 16 line 301, please delete “the” from “… is another the main factors … ”.
11. Page 16, line 309, please move “instead of covering …in one acquisition,” before “the metabolite measurements in this study were…” for clarification.
12. Page 17, line 330, please add “measured” between “right hippocampus” and “by 2D 1HMRS…”.
13. Table 1, in the notes, NPI was mentioned, but not presented in the table, please add. HAMD values were presented in the table; please also describe what it stands for in the notes.
14. Table 3, why the Clinical Dementia Rating (CDR) score was not included? Please include this because patients underwent CDR too (page 6 line 111). Or describe why CDR was not included in the analysis.
15. Figure 2 and 3, since box plots were provided, please also provide mean and standard deviation in the figures.
16. Figure 4, please remake the figure so that the font can be clearer.

·

Basic reporting

Figure 2 A.B.D.E, what MRS are these? are they average spectrum or from one of the participants? May be showing an overlap of all spectra for each group together will give read an overall view of the spectra data.


Minor:

1: P2L43The mI/Cr ratio from the right hippocampus in non-demented elders negatively was correlated with Montreal Cognitive Assessment (MoCA) scores.

was negatively

2. P4L71: However, more and more evidences showed that although ml initially changed in early AD,

should be mI, not ml

3. P8L161: acquisition time is about 6min6s.

6 mins?

4. P9L172 we denote by “partial rs.”

“partial rs”.

Experimental design

The authors state that it used a 2D 1H MRS, but there is no description about how many voxels were collected for each side of hippocampus for each participant, and how do they decide which one value to use among them.

The advantage of 2D vs single voxel MRS is not very clear, except that single voxel MRS can be placed at the bad location with not-pure hippocampus tissue.

It is not clear how many aMCI are APOEε4 carriers, and how many CNs are APOEε4 carriers, and why the authors choose to test the correlation between mI/Cr and MoCA by carriers and non-carries grouping.

Validity of the findings

no comments

Additional comments

1. This is a good paper that presented its effort on providing a window on objectively understand the mechanism between the brain metabolites and their influence factors in aMCI patients.
2. This paper shows the promising contribution of MRS on neuropsychological research and clinic application.
3. The description of some details in English is not very clear.

---

## Round 0.2 · accepted · Accept

Please make an effort to revise the title at proof stage, such as the one suggested by reviewer 3:

Apolipoprotein E genotype and age modify the correlation between cognitive status and hippocampus metabolites measured by a 2D 1HMRS in non-demented elders

Reviewer 1 ·

Basic reporting

No Comments.

Experimental design

No Comments.

Validity of the findings

No Comments.

Additional comments

The authors have revised the manuscript based on my suggestion. I recommend it for publishment.

Reviewer 2 ·

Basic reporting

No Comments

Experimental design

No Comments

Validity of the findings

No Comments

Additional comments

The manuscript has been well-revised. Thank you for the clear response. I have no further suggestion.

Reviewer 3 ·

Basic reporting

Pass

Experimental design

Pass

Validity of the findings

Pass

Additional comments

My previous comments and questions have been addressed except for the title. Please change the title to be clearer. A good example could be:

Apolipoprotein E genotype and age modify the correlation between cognitive status and hippocampus metabolites measured by a 2D 1HMRS in non-demented elders